# The impact of municipal solid waste sorting policy on air pollution: Evidence from Shanghai, China

**Yaopei Wang**[1], **Qingling Shi**[2,3]*

**1** School of Public Economics and Administration, Shanghai University of Finance of Economics, Shanghai, China, **2** The Center for Modern Chinese City Studies, East China Normal University, Shanghai, China, **3** Institute of Urban Development, East China Normal University, Shanghai, China

* qlshi@iud.ecnu.edu.cn

**Data Availability Statement:** The data in this study involved four categories: geographic information, daily air pollution data, daily weather data and some factors included to control for fixed effects. The daily air pollution data and geographic information are from the "Environmental

## Abstract

Municipal solid waste (MSW) sorting not only reduces the total quantity of domestic waste but also has positive effects on urban air quality. In this study, using a careful identification strategy and air quality data at the monitoring station level in Shanghai, we estimate the causal effect of the MSW sorting policy on urban air quality. The results show that after the MSW sorting policy was implemented, the air quality index (AQI), nitrogen dioxide ($NO_2$) and carbon monoxide (CO) decreased significantly by 2.71%, 2.07% and 3.62%, respectively. We also find a positive spillover effect from the Shanghai MSW sorting policy on the air quality of adjacent cities. The implementation of this policy has triggered changes in residents' behaviors. However, the government needs further efforts to maintain the sustainability of MSW sorting policies.

## 1. Introduction

Domestic waste represents an inevitable byproduct of human activity and a major crisis for communities across the globe. There were 2.01 billion tons of municipal solid waste (MSW) produced in 2016, more than 33% of which was incinerated or landfilled [1]. Moreover, MSW incineration is accompanied by several negative externalities. First, the waste incineration process produces large-scale carbon oxides, which are one of the greenhouse gases. In 2016, an equivalent 5% of global carbon emissions were generated from waste incineration [1]. Moreover, the waste incineration process also pumps nitrogen oxides (NOx), sulfur dioxide ($SO_2$), CO and inhalable particulate matter (PM) into the atmosphere, which leads to poisonous fog and acid rain [2]. These acidic gases, such as $SO_2$ and $NO_x$, form acid rain and photochemical smog, causing serious damage to the human body [3].

However, if treated effectively, MSW could be greatly reduced and does less harm to the environment and human health. According to statistics, approximately 44% of domestic mixed solid waste is kitchen solid waste, which can be separated from mixed solid waste and reduced in situ or via anaerobic digestion [1]. These recycling behaviors will directly reduce the total amount of domestic waste that needs to be incinerated and further benefit the environment and human health.

Monitoring of China"(Environmental Monitoring of China: http://www.cnemc.cn/sssj/). Daily weather data are from "China weather forecast platform" (China weather forecast platform: http://www.weather.com.cn/). The datasets used and/or analysed during the current study are available from in Supporting information profile.

**Funding:** Financial support from the National Natural Science Fund of China (Grant No.72203063).

**Competing interests:** The authors have declared that no competing interests exist.

As early as 2004, China surpassed the United States to become the country that produces the largest amount of daily MSW. Over the years, the total amount of MSW in China has reached 6 billion tons (www.stats.gov.cn), and more than 60% is incinerated. To release the negative influence of MSW on the environment, the Chinese government promulgated regulations on MSW sorting and chose Shanghai as the first pilot city to implement the mandatory MSW sorting policy from July 1st, 2019. After half a year, by implementing the MSW sorting policy, the average amount of residual waste that needs to be incinerated in Shanghai decreased by 17.5%.

Therefore, in this study, we focus on the questions of whether the MSW sorting policy in Shanghai significantly improves urban air quality and how large the influence will be. Theoretically, the implementation of MSW sorting policies can promote resource recovery and reduce the total amount of incinerated waste, thereby helping to reduce air pollution [4]. To prove this hypothesis, we estimate the causal effect of MSW sorting policies on urban air quality by using daily data on air quality, weather and geographical information. The urban air quality is measured by the air quality index (AQI) and five air pollutant densities: $SO_2$, $NO_2$, CO, particulate matter 10 ($PM_{10}$), and particulate matter 2.5 ($PM_{2.5}$) at the air monitoring station level. We also checked ozone ($O_3$) as a placebo test. The weather and geographic information were used to estimate the distance and wind direction from 10 monitoring points to 8 domestic waste incineration plants in Shanghai. We use wind directions to causally estimate the pollution effect from incineration plants and apply the difference-in-differences (DID) method to obtain indicative results. After that, a parallel trend test was performed to check if our DID results were valid, and some sensitivity tests were estimated to check the robustness of the results.

The research extends two streams of literature. First, there are numerous studies discussing the ex post effects of air pollution on a wide range of outcomes, such as higher crime rates [5], sleeplessness [6], lower student attendance rates [7], worse performance of marathons [8], lower labor productivity [9], and even higher adult mortality when the term is long enough [10]. However, there are a few studies on different ex ante factors that induce air quality degradation. At present, there is rich literature discussing how traffic influences urban air quality [11–13], but less research studies the influence of other sources of air pollution [14–16] and how we can alleviate it [17, 18]. Second, this research tries to enrich the research in the field of MSW management. At present, limited research has estimated the causal effect of MSW sorting on air pollution from an economic perspective. Chen et al. (2022) [4] used daily air quality data at the air monitoring station level to show that MSW incineration significantly affects the air quality in Shanghai. However, they did not estimate the influence of the MSW sorting policy. Taking the example of Shanghai as the first MSW sorting pilot, we develop a clear and clean research design to investigate the positive consequences of the MSW sorting policy on urban air quality by using the DID method. In addition, our results have implications for future research and policy. Air pollution and waste management are two substantial global issues. According to our conclusion, MSW sorting effectively contributes to the release of both of the problems. Therefore, continuing to implement MSW sorting policies and improving waste management systems are good ideas to tackle these two problems simultaneously. Governments need further efforts to inspire MSW sorting behaviors.

There are three main findings. First, the DID regression results show that the AQI, $NO_2$ and CO declined by 2.71%, 2.07% and 3.62%, respectively, after July 1st, 2019. The results of sensitivity tests and placebo tests also support our findings. Second, we find that there are heterogeneities of this effect by different distances to the incineration plants. The closer to the incineration plants, the larger the influence will be. Third, we also find a positive spillover effect from the Shanghai MSW sorting policy to the air quality in the adjacent city of Kunshan.

The AQI, $NO_2$ and CO in Kunshan also declined in small quantities after Shanghai started the MSW sorting policy. Furthermore, we discuss the externalities of the MSW sorting policy: the MSW sorting policy has triggered changes in residents' awareness and behaviors but also causes inconvenience to residents.

The remainder of this paper proceeds as follows. In the following section, we first demonstrate the background of the MSW sorting policy and verify the research hypotheses. Then, section 2.2 outlines the data used in the research and provides descriptions of the data cleaning process and summary statistics. In section 2.3, we present our econometric model and describe the identification assumptions. Section 3 presents our results, section 4 provides a discussion of the externalities, and section 5 concludes.

## 2. Materials and methods

We first summarize the MSW sorting policy background and introduce the research hypotheses. Then, to test the hypotheses, we describe several datasets and explain our pretreatment. Finally, section 2.3 describes the empirical strategies used to estimate the effect of the MSW sorting policy on urban air quality.

### 2.1 Policy background and research hypotheses

To reduce the negative impacts caused by MSW, many countries have adopted various actions since the 1960s. For example, Japan, Germany, the Netherlands and the United States have achieved remarkable results: approximately 43% of domestic waste is reused as materials or energy [19]. In fact, Beijing proposed the concept of MSW sorting as early as 1955. However, MSW sorting in China was always just an initiative until Shanghai became the first pilot city to implement a mandatory MSW sorting policy. Shanghai, the largest city in China, issued the Regulations of Shanghai municipality on the Administration of Domestic Waste on January 31[st], 2019, and implemented a mandatory MSW sorting policy starting on July 1[st], 2019.

According to the MSW sorting policy in Shanghai, domestic waste is divided into four categories: recyclable waste, hazardous waste, kitchen solid waste and residual waste. Among them, recyclable materials and hazardous waste are easy to distinguish. Recyclable materials are materials that can be recycled, including waste paper, waste plastic, waste glass products, waste metal, etc. Hazardous waste is a waste with properties that make it dangerous or capable of having a harmful effect on human health or the environment. Normally, hazardous waste includes waste batteries, waste lamps, waste drugs, waste paints and containers.

What is new to residents is how to divide kitchen solid waste and residual waste. The kitchen solid waste basically includes all kinds of food, melon peels, fruit cores, flowers, green plants and other perishable wastes. However, the kitchen solid waste should exclude hard fruit shells such as coconut shell, durian core, pineapple honey core, etc. Although they are degradable, they are classified as residual waste because they are not suitable for kitchen solid waste treatment processes. The residual waste is waste in addition to the above three kinds of waste. Figs 1 and 2 show the differences between the MSW treatment processes before and after the MSW sorting policy [20]. The main change in the two MSW treatment processes is that MSW sorting behavior helps to divide kitchen solid waste from residual waste. This will significantly reduce the total amount of MSW going under incineration [21]. By January [1,] 2020, the average daily output of kitchen solid waste in Shanghai was 9006 tons, and the average daily output of residual waste was reduced by more than 2000 tons after the MSW sorting policy in Shanghai.

Based on the above background and evidence, there is a significant reduction in residual waste after implementing the MSW policy. Therefore, the amount of waste that is treated in

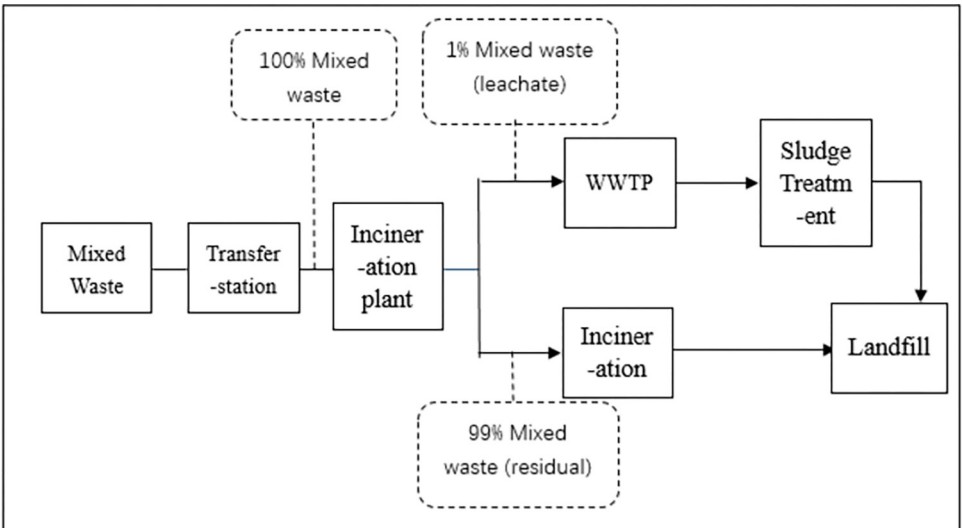

**Fig 1. MSW treatment process before the MSW policy (before July 1ˢᵗ, 2019).** Note: WWTP means "Waste water treatment plants".

the incineration process is decreased. This may reduce the generation of air pollutants and improve urban air quality. we hypothesize as follows:

**H1:** The MSW sorting policy in Shanghai will reduce the air pollutants generated by waste incineration plants and significantly improve the air quality in Shanghai.

Due to the fluidity of air, the influence on air quality will vary from different distances to waste incineration plants [22]. In the meantime, there may be some spillover effect on the air quality in adjacent cities. Consequently, we hypothesized the following:

**H2:** The degree of influence on air quality will be affected by the distances to waste incineration plants. The closer to waste incineration plants, the larger the influence will be.

**H3:** There will be a positive spillover effect on the air quality in adjacent cities.

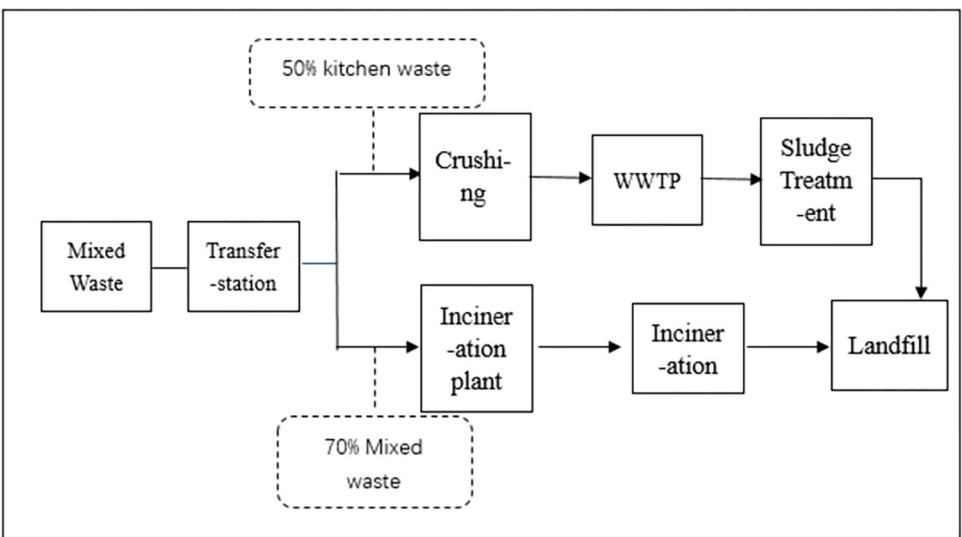

**Fig 2. MSW treatment process after the MSW policy (After July 1ˢᵗ, 2019).**

## 2.2 Data source and description

To carefully test our three hypotheses, we built our panel dataset from four categories: daily weather data, daily air quality data, geographic information and some factors included to control for fixed effects.

The daily weather data are from the official website: *China weather forecast platform* (http://www.weather.com.cn/), including weather data for districts and counties in each city. The specific indicators include maximum temperature (*Tem_h*), minimum temperature (*Tem_l*), wind force (*Windf*), wind direction and rainfall. According to the intensity of rainfall, we generate the ordinal variable *rainfall*. If there is no rain, the *rainfall* value is 0. The variable *rainfall* values ranged from 1 to 5 for light rain (0.1–9.9 mm/d), medium rain (10–24.9 mm/d), heavy rain (25–49.9 mm/d), torrential rain (50–99.9 mm/d), and downpour (100–250 mm/d), respectively.

The daily air quality data at the air monitoring station level were derived from the Ministry of Environmental Protection of China (http://www.cnemc.cn/sssj/). Our data source is the same as that in certain previous studies related to air pollution in China [23, 24]. This ministry started publishing the six daily air quality data from 2014: AQI, $SO_2$, $NO_2$, CO, $PM_{10}$, $PM_{2.5}$ and $O_3$. Among them, AQI, $SO_2$, $NO_2$, CO, $PM_{10}$ and $PM_{2.5}$ may be affected by solid waste incineration, while $O_3$ is not directly related to waste incineration. We also analyze the impact of MSW sorting on $O_3$ as a placebo test.

Geographic information plays an important role in estimating the causal effect of MSW sorting policies on urban air quality. To accurately detect the changes in air quality before and after the implementation of the policy, we carefully build links between urban air quality and domestic waste incineration. We use the daily wind direction as well as the locations of domestic waste incineration plants and air monitoring stations to determine whether the air monitoring stations are downwind of waste incineration plants. If downwind, then the air quality is directly influenced by domestic waste incineration. At some dates in our dataset, air monitoring stations are downwind of waste incineration plants, and the air quality data of these air monitoring stations are in the treatment group; otherwise, the air quality data are in the control group. There are 8 domestic waste incineration plants and 10 air monitoring stations in Shanghai. Their locations are shown in Fig 3.

Using the ArcGIS tool, the distances between each air monitoring station and the domestic waste incineration plants are calculated. The weather dataset provides the daily wind direction in each district of Shanghai, and the wind direction is grouped into 8 directions: east, southeast, south, southwest, west, northwest, and north. We use the wind directions and relative locations of air monitoring stations (stars in Fig 3) and domestic waste incineration plants (dots in Fig 3) to estimate if an air monitoring station is located downwind to a domestic waste incineration plant at a certain date. For example, if the wind direction is southeast today, the downwind means air monitoring stations are located in the southeast direction of a domestic waste incineration. The variable *Downwind1* shows the total number of domestic waste incineration plants to which each air detection point was located downstream, which ranges from 0 to 8. We also generate *Downwind2* to estimate how distance affects the influences caused by domestic waste incineration. These are the two main variables used to test hypothesis 1 and hypothesis 2. The descriptive statistics of the variables are shown in Table 1.

We also use the air quality data of the two nearest air monitoring stations in adjacent city Kunshan to test the spillover effect. There is one domestic waste incineration plant only 1 kilometer from the border of Kunshan city. The two stars in the gray area in Fig 3 show the locations of the air monitoring stations in Kunshan. We use them to test hypothesis 3: if there is a spillover effect from the MSW sorting policy in Shanghai to the air quality in the adjacent city Kunshan (Table 2).

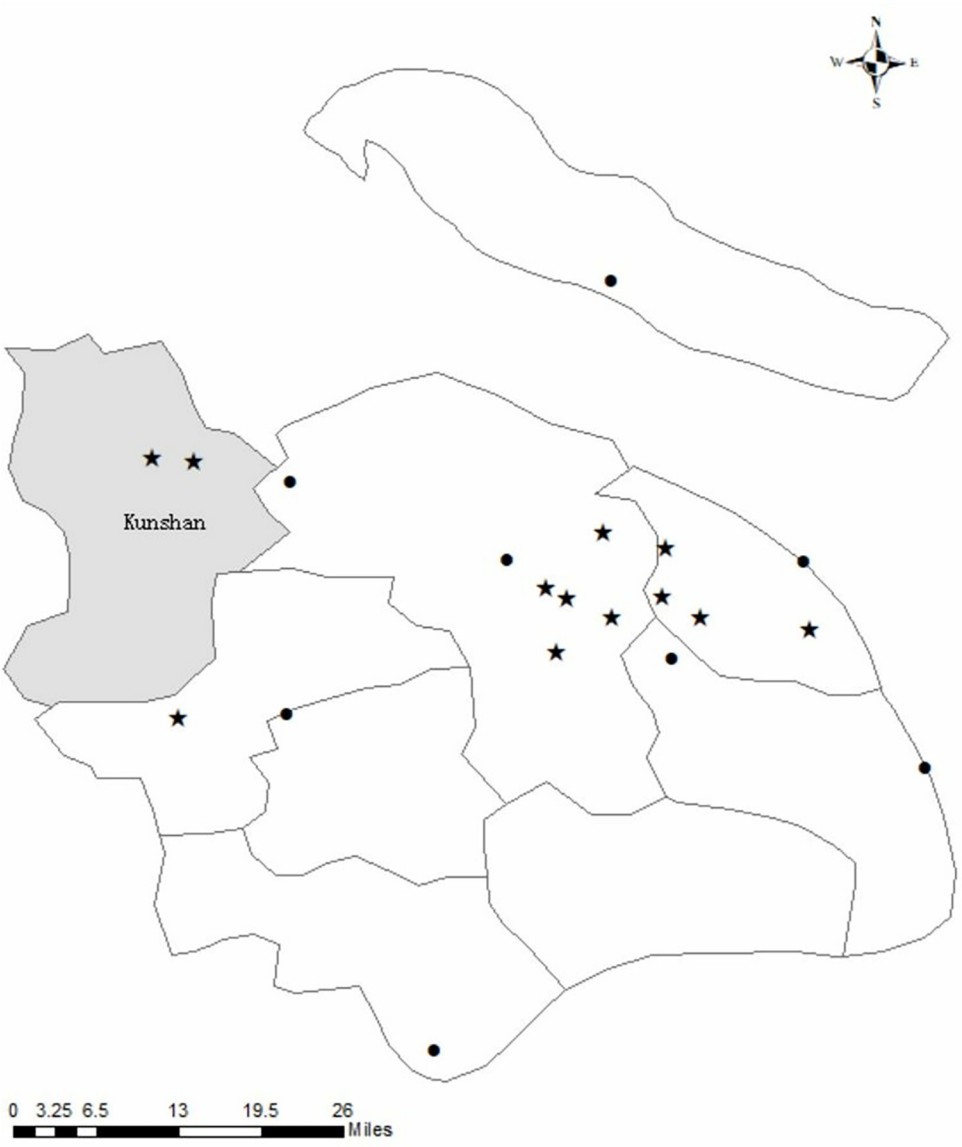

**Fig 3. The locations of air monitoring stations (stars) and domestic waste incineration plants (dots).** Note: Base map and data from OpenStreetMap and OpenStreetMap Foundation, original copyright: https://www.openstreetmap. org/copyright. The stars mark the locations of air monitoring stations, and the dots mark the locations of domestic waste incineration plants in Shanghai.

By matching air quality data with weather data, daily panel data were obtained. We summarize 10 air monitoring stations located in Shanghai from 1st January to 31st December 2019 and 2 air monitoring stations located in Kunshan city from 1st April to 30th September 2019. Furthermore, holidays, weekday fixed effects, month fixed effects and air monitoring point fixed effects are also taken into consideration.

## 2.3 Empirical strategy

As mentioned in section 2.2, we estimate whether the air monitoring station is downwind of the domestic waste incineration plants to build the treatment and control groups in our DID regression. This helps us deal with potential endogeneities and estimate the casual influence of

**Table 1. Descriptive statistics of the main variables (Shanghai).**

| Variable | Obs | Mean | Std. Dev. | Min | Max |
|---|---|---|---|---|---|
| $CO$ $(mg/m^3)$ | 3,485 | 0.723 | 0.364 | 0.16 | 3.5 |
| $NO_2$ $(mg/m^3)$ | 3,485 | 40.623 | 19.734 | 2.57 | 137.521 |
| $O_3$ $(mg/m^3)$ | 3,485 | 66.661 | 31.390 | 3.04 | 192.257 |
| $PM_{10}$ $(mg/m^3)$ | 3,485 | 55.826 | 41.970 | 0 | 305.044 |
| $PM_{2.5}$ $(mg/m^3)$ | 3,485 | 38.086 | 26.260 | 0 | 191.826 |
| $SO_2$ $(mg/m^3)$ | 3,485 | 8.536 | 6.341 | 1 | 75 |
| $AQI$ | 3,485 | 58.721 | 34.001 | 0 | 241.818 |
| $Tem\_h$ (°C) | 3,485 | 21.360 | 8.559 | 4 | 37 |
| $Tem\_l$ (°C) | 3,485 | 14.642 | 7.966 | -3 | 30 |
| $Weekday$ | 3,485 | 3.960 | 2.002 | 1 | 7 |
| $Holidays$ | 3,485 | 0.0767 | 0.266 | 0 | 1 |
| $Windf$ | 3,485 | 2.621 | 0.684 | 1 | 7 |
| $Distance$ (km) | 3,485 | 9.543 | 2.695 | 5.358 | 14.220 |
| $rainfall$ | 3,485 | 0.576 | 0.771 | 0 | 4 |
| $Downwind1$ | 3,485 | 1.967 | 1.160 | 0 | 6 |
| $Downwind2$ | 3,485 | 0.083 | 0.049 | 0 | 0.219 |

MSW policy on urban air quality. A nascent body of literature exploits variations in wind directions to causally estimate pollution's effect [25–28]. We use *Downwind1* to denote the numbers of domestic waste incineration plants that each air monitoring station was located downstream to at a certain date. As such, the primary model specification that we deploy for the majority of our analyses takes the following form:

$$Y_{air\ i,t} = \alpha_0 + \beta_1 Downwid1_{i,t} \times policy_{i,t} + \beta_2 Downwid1_{i,t} + \beta_3 policy_{i,t} + \gamma X_{weather\ t} + \tau_{weekday}$$
$$+ \pi_{monitor} + \mu_{month} + \varepsilon \tag{1}$$

We adopt a semi-log model to empirically test the impact of the MSW sorting policy on daily air quality. where $Y_{air\ i,t}$ denotes the logarithm of the $SO_2$, $NO_2$, $CO$, $PM_{10}$, $PM_{2.5}$, $O_3$ and $AQI$ at the air monitoring station level. The *policy* is a time-related dummy variable that equals

**Table 2. Descriptive statistics of the main variables (Kunshan).**

| Variable | Obs | Mean | Std. Dev. | Min | Max |
|---|---|---|---|---|---|
| $CO$ $(mg/m^3)$ | 347 | 0.706 | 0.195 | 0.3 | 1.7 |
| $NO_2$ $(mg/m^3)$ | 347 | 27.282 | 12.344 | 4 | 97 |
| $O_3$ $(mg/m^3)$ | 347 | 151.253 | 60.788 | 38 | 391 |
| $PM_{10}$ $(mg/m^3)$ | 347 | 50.550 | 20.780 | 9 | 127 |
| $PM_{2.5}$ $(mg/m^3)$ | 347 | 26.371 | 14.639 | 4 | 161 |
| $SO_2$ $(mg/m^3)$ | 347 | 9.648 | 17.220 | 3 | 322 |
| $AQI$ | 347 | 47.907 | 20.901 | 8 | 157 |
| $Tem\_h$ (°C) | 350 | 27.857 | 4.822 | 14 | 37 |
| $Tem\_l$ (°C) | 350 | 20.628 | 5.285 | 6 | 30 |
| $Weekday$ | 350 | 3.948 | 2.020 | 1 | 7 |
| $Holidays$ | 350 | 0.074 | 0.262 | 0 | 1 |
| $Windf$ | 350 | 2.428 | 0.618 | 1 | 5 |
| $rainfall$ | 350 | 0.622 | 0.930 | 0 | 4 |
| $Downwind$ | 350 | 0.462 | 0.499 | 0 | 1 |

1 when the MSW sorting policy is implemented and equals 0 otherwise. The cross variable *Downwind1×policy* is the DID item used to causally estimate the policy effect on air pollution. $X_{weather}$ is a vector of daily weather variables, including *Tem_h*, *Tem_l*, *Windf*, and *rainfall*. There are two time fixed effects: $\tau_{weekday}$ to control for differences between each weekday and $\mu_{month}$ to capture the monthly time effect. We also include $\pi_{monitor}$ to control individual effects. The error term $\varepsilon$ is clustered by air monitoring points to allow for autocorrelations.

It is notable that in general, all of the air monitoring stations are more than 5 kilometers away from the nearest waste incineration plant, and half of them are 10 kilometers away (see Fig 3). This means that residents approximately 5 kilometers will benefit from the implemented policies if MSW sorting behavior does have a significant influence on air quality. Additionally, it is notable that the diffusion scale and radius of the plant plum decrease the estimated value, which means that the effects of the MSW sorting policy should be larger than we estimated.

To better understand the robustness of the baseline results and estimate heterogeneities in distance, we regress Eq (1) with different distance limitations. The weighted downwind variable (*downwind2*) was constructed to measure the influence of distances from the domestic waste incinerations and test hypothesis 2:

$$Downwind2_{i,t} = \sum_{j=1}^{n} \left( {}^{1}/_{Distance_{j,i}} \times Downwid1_{j,i,t} \right) \qquad (2)$$

where *Downwind2*$_{i,t}$ denotes the distance weighted downwind variable of air monitoring station $i$ on day $t$ and *Distance*$_{j,i}$ represents the distance from air monitoring station $i$ to domestic waste incineration plant $j$. Therefore, the regression is as follows:

$$Y_{air i,t} = \alpha_0 + \beta_1 downwind2_{i,t} \times policy_{i,t} + \beta_2 downwind2_{i,t} + \beta_3 policy_{i,t} + \gamma X_{weather\,t} + \tau_{weekay}$$
$$+ \pi_{monitor} + \mu_{month} + \varepsilon \qquad (3)$$

In this DID regression, the reciprocal of the distance is taken as the weight of the wind direction: *Downwind2*$_{i,t}$ was used to capture the combined effect of domestic waste incineration with distance. The economic explanation of parameter $\beta_1$ is the heterogeneity in the MSW sorting policy effect with distance.

After obtaining the baseline results of the MSW sorting policy, we check the parallel trend test of the baseline and perform robustness tests, including a sensitivity test and placebo test. The parallel trend test helps to ensure that DID regression is effective, and robustness tests check whether the basic results can be used in a wider scope. The spillover effect of the municipal solid waste sorting policy on the air quality of Kunshan city is also explored by using Eq (1).

## 3. Results

This section presents our empirical results. The following subsection presents the baseline findings by applying DID regression. Subsection 3.2 presents the results of the common trend test. Subsection 3.3 presents a series of robustness checks, including sensitivity tests and placebo tests. Finally, we test the spillover effect of the Shanghai MSW sorting policy on the air quality of the adjacent city of Kunshan.

### 3.1 Baseline findings

Tables 3 and 4 report the results of regressions (1) and (3) using pooled data in Shanghai and prove hypothesis 1 and hypothesis 2. The coefficients of the two variables and policy dummy are statistically significant and stable when the dependent variables are AQI, $NO_2$ and CO across the two tables. Table 3 shows that after the MSW sorting policy, the AQI, $NO_2$ and CO

**Table 3. Difference-in-differences panel data regression results (Downwind1).**

|  | AQI | PM$_{2.5}$ | PM$_{10}$ | NO$_2$ | SO$_2$ | CO | O$_3$ |
|---|---|---|---|---|---|---|---|
| *Downwind1×policy* | -0.0271** | -0.0259 * | -0.0085 | -0.0207** | 0.0113 | -0.0362*** | -0.0102 |
|  | (0.030) | (0.085) | (0.485) | (0.031) | (0.275) | (0.001) | (0.262) |
| *Downwind1* | 0.0106 | 0.0116 | 0.0109 | 0.0143* | -0.0081 | 0.0087 | -0.0084 |
|  | (0.275) | (0.323) | (0.272) | (0.053) | (0.318) | (0.287) | (0.341) |
| *policy* | 0.0638 | 0.1423*** | -0.0872** | 0.2204*** | 0.0120 | 0.1192*** | -0.0084 |
|  | (0.138) | (0.006) | (0.043) | (0.000) | (0.738) | (0.001) | (0.832) |
| *Holidays* | -0.1312*** | -0.2481*** | -0.1082*** | -0.3097*** | -0.0234 | -0.0826*** | 0.0031 |
|  | (0.000) | (0.000) | (0.000) | (0.000) | (0.329) | (0.001) | (0.906) |
| *Rain* | -0.0223** | -0.0349*** | -0.1195*** | 0.0048 | 0.0120 | 0.0123 | -1.1183*** |
|  | (0.038) | (0.007) | (0.000) | (0.560) | (0.738) | (0.180) | (0.000) |
| *Wind_force* | 0.0067 | -0.3033*** | -0.1679*** | -0.3367*** | -0.1165*** | -0.1459*** | 0.0064 |
|  | (0.562) | (0.000) | (0.000) | (0.000) | (0.000) | (0.000) | (0.539) |
| *Tem_h* | 0.0158*** | 0.0501*** | 0.0593*** | 0.0195*** | 0.0155*** | 0.0189*** | 0.0289 |
|  | (0.000) | (0.000) | (0.000) | (0.000) | (0.000) | (0.000) | (0.000) |
| *Tem_l* | -0.0182*** | -0.0413*** | -0.0563*** | -0.0167*** | -0.0202*** | -0.0120*** | -0.0428 |
|  | (0.000) | (0.000) | (0.000) | (0.000) | (0.000) | (0.000) | (0.000) |
| *Weekday FE* | √ | √ | √ | √ | √ | √ | √ |
| *Month FE* | √ | √ | √ | √ | √ | √ | √ |
| *Station FE* | √ | √ | √ | √ | √ | √ | √ |
| Obs | 3,410 | 3,482 | 3,448 | 3,485 | 3,485 | 3,485 | 3,485 |
| Adjust R$^2$ | 0.2964 | 0.3603 | 0.5459 | 0.5794 | 0.5760 | 0.3154 | 0.5382 |

Note: Dependent variables are natural logarithms of air quality data. All models include a month dummy variable and air monitoring station fixed effects. * indicates significance at the 10% level. ** indicates significance at the 5% level. *** indicates significance at the 1% level. The observation is less than 3485 because some values of air quality data are recorded as 0.

decreased by 2.71%, 2.07% and 3.62%, respectively. Table 4 suggests that there are heterogeneous influences with distance: the nearer a monitoring site is to an incineration plant, the larger the decrease in the AQI, NO$_2$ and CO will be.

In contrast, the pollution values of PM$_{2.5}$, PM$_{10}$, and SO$_2$ did not show significant changes after the MSW sorting policy was implemented. The explanation of these results is that MSW sorting behaviors mainly reduce the total amount of mixed waste by removing kitchen solid waste, but recyclable waste and hazardous waste occupy small proportions. Therefore, the organic matter and wastewater are significantly reduced in the incineration process. Most organic matter is composed of carbides and nitrides. Combustion is not sufficient in the presence of water, resulting in NO$_x$ and CO pollution. From the source of reduced MSW incineration, PM$_{2.5}$, PM$_{10}$, and SO$_2$ should not be significantly affected. There may be a slight decrease in PM$_{2.5}$, PM$_{10}$, and SO$_2$ in the area that is very close to incineration plants. Due to the limitation of the dataset, we could not prove these.

These results also show that urban air quality is closely related to weather conditions and living behaviors. We find that PM$_{2.5}$ and PM$_{10}$ significantly decline on rainy days and that a stronger wind force reduces the pollution of PM$_{2.5}$, PM$_{10}$, NO$_2$, CO and SO$_2$. The public holiday effects are also controlled for in the regression. PM$_{2.5}$, AQI, PM$_{10}$, NO$_2$, CO and SO$_2$ decline significantly on holidays.

It is worth mentioning that we also included O$_3$ in our regression as a placebo test of the influence of MSW sorting. The O$_3$ content in the atmosphere is not related to domestic waste

**Table 4. Difference-in-differences panel data regression results (Downwind2).**

|  | AQI | PM$_{2.5}$ | PM$_{10}$ | NO$_2$ | SO$_2$ | CO | O$_3$ |
|---|---|---|---|---|---|---|---|
| *Downwind2×policy* | -0.9879*** | -0.3250 | -0.0072 | -0.3824* | 0.1731 | -0.3997* | -0.3792 |
|  | (0.001) | (0.364) | (0.980) | (0.092) | (0.484) | (0.087) | (0.162) |
| *Downwind2* | 0.4082* | 0.1372 | 0.3311 | 0.2483 | -0.2233 | -0.1736 | -0.1897 |
|  | (0.054) | (0.579) | (0.121) | (0.127) | (0.208) | (0.337) | (0.329) |
| *policy* | 0.0926** | 0.1170** | -0.1026** | 0.2095*** | 0.0198 | 0.0798** | -0.0037 |
|  | (0.031) | (0.025) | (0.015) | (0.000) | (0.582) | (0.030) | (0.925) |
| *Holiday* | -0.1303*** | -0.2488*** | -0.1056*** | -0.3013*** | -0.0230 | -0.0772*** | 0.0043 |
|  | (0.000) | (0.000) | (0.000) | (0.000) | (0.357) | (0.002) | (0.871) |
| *Weather* | √ | √ | √ | √ | √ | √ | √ |
| *Weekday FE* | √ | √ | √ | √ | √ | √ | √ |
| *Month FE* | √ | √ | √ | √ | √ | √ | √ |
| *Station FE* | √ | √ | √ | √ | √ | √ | √ |
| Obs | 3,410 | 3,482 | 3,448 | 3,485 | 3,485 | 3,485 | 3,485 |
| R$^2$ | 0.2977 | 0.3598 | 0.5463 | 0.5792 | 0.5761 | 0.3147 | 0.5385 |

Note: Dependent variables are natural logarithms of air quality data. All models include a month dummy variable and air monitoring station fixed effects.

* indicates significance at the 10% level.

** indicates significance at the 5% level.

*** indicates significance at the 1% level. The observation is less than 3485 because some values of air quality data are recorded as 0.

incineration. The DID variables of O$_3$ in Tables 3 and 4 are not significant, which also supports hypothesis 1.

## 3.2 Parallel trend test

To check the validity of the DID results in Tables 3 and 4, we perform the parallel trend test to ensure that there are no significant differences between the treated group and the control group before the MSW policy. We use Jacobson's method (1993) [29] to test the parallel trend and the event analysis framework to evaluate the dynamic effect of the policy:

$$Y_{air\ i,t} = \alpha_0 + \sum_{k=-3,k\neq -1}^{3} \beta_i Downwind1_{i,t} \times month\_dunmmy + \gamma X_{weather\ t} + \tau_{week} + \pi_{monitor} + \mu_{month}$$
$$+ \varepsilon \tag{4}$$

where k = -1 is two months before the policy, and we use it as the baseline. *Post_1* with a value of 1 represents the first two months after the policy was implemented. The meaning of the remaining *month_dunmmy* can be deduced by this regular pattern.

When k>0, $\beta_i$ reflects the impact of the policy in the month of policy implementation and subsequent months. When k≤0, $\beta_i$ captures whether the impact of policy anticipation exists in the month before policy implementation, and the definition of other variables remains unchanged. Therefore, it is necessary to test whether the effect $\beta_i$ produced by the policy is significantly different from 0 before the implementation of the policy (k≤0). If it is significantly different from 0, then the policy was expected by the individual in advance, and there is a trend in advance. Thus, the regression coefficient of the DID method has no causal explanation. The results in Table 5 show that the DID models in section 3.1 meet the requirement of the parallel trend hypothesis. Therefore, the baseline results in section 3.1 are indicative.

**Table 5. Parallel trend test.**

|  | AQI | PM$_{2.5}$ | PM$_{10}$ | NO$_2$ | SO$_2$ | CO | O$_3$ |
|---|---|---|---|---|---|---|---|
| *pre_3* | 0.3459 | -0.3465 | 0.2881 | -0.0005 | -0.2891 | -0.7002 | 0.6147 |
|  | (0.497) | (0.575) | (0.581) | (0.999) | (0.393) | (0.109) | (0.189) |
| *pre_2* | 0.6312 | 0.4176 | -0.2925 | 0.4670 | -0.1812 | 1.1117* | 0.5013 |
|  | (0.213) | (0.499) | (0.565) | (0.233) | (0.898) | (0.071) | (0.284) |
| *post_1* | -0.5854* | -0.4006 | -0.221 | -0.6459** | -0.1209 | -0.6254* | 0.5637* |
|  | (0.099) | (0.370) | (0.967) | (0.023) | (0.741) | (0.096) | (0.052) |
| *post_2* | -0.7662** | 0.5904 | 1.160*** | 0.2049 | 0.6631** | -0.5999* | 0.4327 |
|  | (0.046) | (0.209) | (0.002) | (0.286) | (0.038) | (0.067) | (0.137) |
| *post_3* | -0.4508 | -0.6527 | -0.1081 | 0.0420 | 0.0874 | -0.2438 | -0.1150 |
|  | (0.209) | (0.129) | (0.759) | (0.877) | (0.768) | (0.420) | (0.679) |
| Weather | √ | √ | √ | √ | √ | √ | √ |
| Weekday FE | √ | √ | √ | √ | √ | √ | √ |
| Month FE | √ | √ | √ | √ | √ | √ | √ |
| Station FE | √ | √ | √ | √ | √ | √ | √ |
| Obs | 3,410 | 3,482 | 3,448 | 3,485 | 3,485 | 3,485 | 3,485 |
| R$^2$ | 0.3042 | 0.3668 | 0.5514 | 0.5836 | 0.5823 | 0.3240 | 0.5442 |

Note: Dependent variables are natural logarithms of air quality data. All models include a month dummy variable and air monitoring station fixed effects.

* indicates significance at the 10% level.

** indicates significance at the 5% level.

*** indicates significance at the 1% level. The observation is less than 3485 because some values of air quality data are recorded as 0.

## 3.3 Robustness tests

To further illustrate the rationality of the empirical results, this subsection takes a series of robustness tests. First, we narrowed down the period from 12 months to 6 months to see if the results were sensitive to different time periods. Second, we perform another placebo test by assuming the date of the policy announcement, 1$^{st}$ February, as the policy dummy and rerun the DID regressions.

Table 6 shows that our baseline results are not sensitive to time period. The AQI, NO$_2$ and CO are significantly reduced at the 1% confidence interval level. It is interesting that the AQI, NO$_2$ and CO declined more significantly when the time period was narrowed. Combined with the parallel trend test in Table 5, it seems that the effects of MSW policy become weaker in the long term. The reason may be that executive power has become increasingly weaker over time. We raise this possibility here and will investigate it further in a future study. The placebo test results in Table 7 show that the partial coefficients of the AQI, NO$_2$ and CO are not significant when the policy implementation time is incorrectly set, indicating that there is no expected effect.

In summary, the above results show that the empirical analysis is robust and the empirical results are effective.

## 3.4 Spillover effect

As described above, one of the largest domestic waste incineration plants in Shanghai is very close to Kunshan city. Therefore, when Kunshan is located downwind of Shanghai, waste incineration plants may affect the air quality of Kunshan city. Consequently, we further examine the spillover effects of the MSW sorting policy on air quality in Kunshan to test hypothesis 3.

**Table 6. Sensitivity test (April to September 2019).**

|  | AQI | PM$_{2.5}$ | PM$_{10}$ | NO$_2$ | SO$_2$ | CO | O$_3$ |
|---|---|---|---|---|---|---|---|
| *Downwind2×policy* | -0.9173** | -0.1557 | -0.0476 | -0.9894*** | -0.0130 | -0.8544*** | -1.6294 |
|  | (0.014) | (0.687) | (0.887) | (0.001) | (0.968) | (0.003) | (0.184) |
| *Downwind2* | 0.3692 | -0.1364 | -0.1073 | 0.1094 | -0.4694** | 0.1066 | 0.5174* |
|  | (0.158) | (0.613) | (0.648) | (0.585) | (0.042) | (0.597) | (0.052) |
| *policy* | -0.2164*** | -0.5891*** | -0.4263 | -0.3049*** | 0.1345*** | -0.1500*** | 0.2687*** |
|  | (0.000) | (0.000) | (0.000) | (0.000) | (0.005) | (0.000) | (0.000) |
| Weather | √ | √ | √ | √ | √ | √ | √ |
| Weekday FE | √ | √ | √ | √ | √ | √ | √ |
| Month FE | √ | √ | √ | √ | √ | √ | √ |
| Station FE | √ | √ | √ | √ | √ | √ | √ |
| Obs | 1,749 | 1,749 | 1,749 | 1,749 | 1,749 | 1,749 | 1,749 |
| R$^2$ | 0.1679 | 0.5048 | 0.5326 | 0.5517 | 0.2292 | 0.2845 | 0.3057 |

Note: Dependent variables are natural logarithms of air quality data. All models include a month dummy variable and air monitoring station fixed effects. * indicates significance at the 10% level. ** indicates significance at the 5% level.

*** indicates significance at the 1% level.

Table 8 presents the DID regression results of two air monitoring stations in Kunshan city from April 1$^{st}$ to September 30$^{th}$. The AQI, CO and NO$_2$ significantly declined after the Shanghai MSW sorting policy was implemented, which means that the policy has a positive effect on the air quality in Kunshan city, which proves hypothesis 3. It is worth mentioning that the significance of the AQI, CO and NO$_2$ are weaker in Kunshan, which is consistent with hypothesis 2: the influence will decline with farther distance.

## 4. Discussion

The above results suggest that MSW sorting policy has significant effects on air quality, especially AQI, NO$_2$ and CO. In this section, we discuss some externalities caused by the MSW sorting policy.

**Table 7. Placebo test (1st Feb 2019 as the placebo policy dummy).**

|  | AQI | PM$_{2.5}$ | PM$_{10}$ | NO$_2$ | SO$_2$ | CO | O$_3$ |
|---|---|---|---|---|---|---|---|
| *Downwind2×placebo* | -0.0711 | -0.1294 | -0.4272*** | 0.0215 | -0.2077 | 0.2107* | 0.4127*** |
|  | (0.649) | (0.519) | (0.009) | (0.872) | (0.118) | (0.089) | (0.000) |
| *Downwind2* | 0.6468** | 0.1660 | 0.5345* | -0.1093 | -0.0497 | -0.2472 | -0.679*** |
|  | (0.017) | (0.613) | (0.060) | (0.635) | (0.828) | (0.324) | (0.000) |
| *placebo* | -0.0602 *** | -0.1689*** | -0.0352 | -0.1680*** | -0.0392** | -0.0883*** | 0.1777*** |
|  | (0.000) | (0.000) | (0.134) | (0.000) | (0.041) | (0.000) | (0.000) |
| Weather | √ | √ | √ | √ | √ | √ | √ |
| Weekday FE | √ | √ | √ | √ | √ | √ | √ |
| Month FE | √ | √ | √ | √ | √ | √ | √ |
| Station FE | √ | √ | √ | √ | √ | √ | √ |
| Obs | 1,478 | 1,478 | 1,478 | 1,478 | 1,478 | 1,478 | 1,478 |
| R$^2$ | 0.1912 | 0.1799 | 0.3958 | 0.3909 | 0.5407 | 0.1545 | 0.5159 |

Note: Dependent variables are natural logarithms of air quality data. All models include a month dummy variable and air monitoring station fixed effects. * indicates significance at the 10% level. ** indicates significance at the 5% level.

*** indicates significance at the 1% level.

**Table 8. Difference-in-differences panel data regression results (spillover).**

|  | AQI | PM$_{2.5}$ | PM$_{10}$ | NO$_2$ | SO$_2$ | CO | O$_3$ |
|---|---|---|---|---|---|---|---|
| *Downwind×policy* | -0.3239*** | -0.1616 | -0.0055 | -0.1840* | -0.0415 | -0.1325* | -0.1187 |
|  | (0.003) | (0.199) | (0.962) | (0.088) | (0.747) | (0.070) | (0.303) |
| *Downwind* | 0.0377 | 0.0108 | 0.1091 | 0.1127** | 0.0352 | -0.0845*** | 0.1778** |
|  | (0.425) | (0.907) | (0.117) | (0.017) | (0.626) | (0.008) | (0.036) |
| *policy* | -0.3063*** | -0.5341 | -0.6730*** | -0.3583*** | 0.2219 | -0.1381** | 0.0909 |
|  | (0.001) | (0.106) | (0.006) | (0.000) | (0.381) | (0.031) | (0.757) |
| *Holiday* | -0.0675 | -0.0988 | -0.0224 | -0.2268*** | 0.0935 | 0.0035 | -0.1776 |
|  | (0.398) | (0.542) | (0.851) | (0.004) | (0.454) | (0.948) | (0.222) |
| *Weather* | √ | √ | √ | √ | √ | √ | √ |
| *Weekday FE* | √ | √ | √ | √ | √ | √ | √ |
| *Month FE* | √ | √ | √ | √ | √ | √ | √ |
| *Station FE* | √ | √ | √ | √ | √ | √ | √ |
| Obs | 347 | 347 | 347 | 347 | 347 | 347 | 347 |
| R$^2$ | 0.4223 | 0.4730 | 0.2588 | 0.4708 | 0.1920 | 0.3466 | 0.1866 |

Note: Dependent variables are natural logarithms of air quality data. All models include a month dummy variable and air monitoring station fixed effects.

* indicates significance at the 10% level.

** indicates significance at the 5% level.

*** indicates significance at the 1% level.

First, the MSW sorting policy, which contributes to urban air quality, will increase social welfare. Research on health losses caused by urban air pollution has a long history [30]. For example, Chen and Chen (2021) [31] found that in China, every unit increase in the AQI leads to 10.013 CNY of further health costs. Therefore, the MSW sorting policy produces millions CNY of health economic welfare. Furthermore, reducing air pollution also benefits mental health [7], increases the well-being of residents [32] and increases the birth rate [33]. The value of these positive externalities is incalculable.

Second, the MSW policy in Shanghai has triggered changes in residents' behaviors. Urban citizens in China have started to show high concern about MSW sorting, which imperceptibly influences living habits. Fig 4 shows the number of people who searched for waste bins, added a waste bin to their shopping cart and purchased a waste bin from 1st Feb to 30th Nov 2019. We can easily see an increasing trend in the numbers after policy implementation, which could help to reveal the MSW sorting behavior of residents in Shanghai. This change reveals that there is an increased awareness of MSW sorting in residents, which is a positive externality of the policy.

Finally, there are still many immature aspects of the MSW sorting policy in Shanghai, which has caused inconvenience to residents. Wu et al. (2021) [34] used a text mining technique to uncover attitudes from the Chinese public toward MSW sorting policies. They find that although a large proportion of the Chinese public has a positive attitude toward the MSW sorting policy, the proportion of people with negative emotions reached nearly half. Most negative emotions were toward fines, MSW sorting rules, fees, timing of throwing waste, and irregular recycling procedures. Our parallel trend test also shows that the impact of the MSW sorting policy seems to become weaker in the long term. Therefore, to maintain the sustainability of MSW sorting behaviors, the government needs to further improve efficiency and reduce residents' time costs.

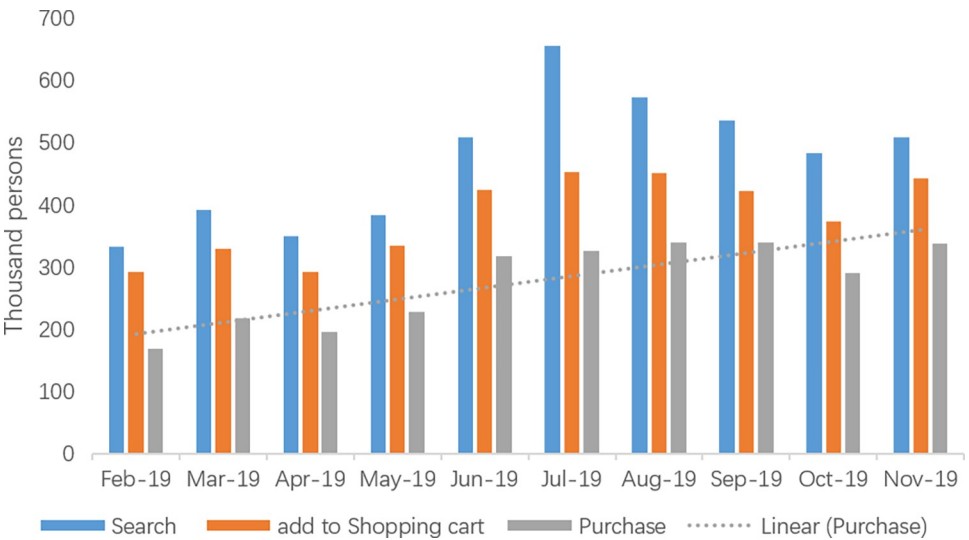

**Fig 4. The number of people who searched for a waste bin, added a waste bin to their shopping cart or purchased a waste bin.**

## 5. Conclusions

Individuals and governments making decisions on waste management will affect human health, productivity, and urban governance. In this study, a careful identification strategy was used to estimate the causal effects of the Shanghai MSW sorting policy on urban air pollution.

The results show that after the MSW sorting policy was implemented, the AQI, $NO_2$ and CO decreased significantly by 2.71%, 2.07% and 3.62%, respectively. This is because the total amount of domestic waste was reduced: kitchen solid waste, which occupied approximately 35% of the total MSW, was separated and put into the crushing process instead. In addition, there is a heterogeneous influence of the distance to waste incineration plants: the closer to waste incineration plants, the larger the influence will be. We also find a positive spillover effect from the MSW sorting policy in Shanghai to the air quality in the adjacent city Kunshan. Although the MSW sorting policy has many positive externalities, there are some negative attitudes from residents. Increasing the efficiency and inspiring the enthusiasm of residents is important to the sustainability of MSW sorting behaviors.

## Supporting information

**S1 File.**
(ZIP)

## Acknowledgments

The Contains information from OpenStreetMap and OpenStreetMap Foundation, which is made available under the Open Database License.

## Author Contributions

**Conceptualization:** Yaopei Wang.

**Data curation:** Yaopei Wang.

**Formal analysis:** Qingling Shi.

**Resources:** Yaopei Wang.

**Supervision:** Qingling Shi.

**Writing – original draft:** Yaopei Wang.

**Writing – review & editing:** Qingling Shi.

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
