## [Decision Letter · Decision Letter 0]

30 Aug 2022

PONE-D-22-13042The Impact of Municipal solid waste sorting policy on Air Pollution: Evidence from Shanghai, ChinaPLOS ONE

Dear Dr. Wang,

Thank you for submitting your manuscript to PLOS ONE. After careful consideration, we feel that it has merit but does not fully meet PLOS ONE’s publication criteria as it currently stands. Therefore, we invite you to submit a revised version of the manuscript that addresses the points raised during the review process.

We look forward to receiving your revised manuscript.

Kind regards,

Ghaffar Ali, PhD

Academic Editor

PLOS ONE

Journal Requirements:

2. We note that Figure 3 in your submission contain map images which may be copyrighted. All PLOS content is published under the Creative Commons Attribution License (CC BY 4.0), which means that the manuscript, images, and Supporting Information files will be freely available online, and any third party is permitted to access, download, copy, distribute, and use these materials in any way, even commercially, with proper attribution. For these reasons, we cannot publish previously copyrighted maps or satellite images created using proprietary data, such as Google software (Google Maps, Street View, and Earth). For more information, see our copyright guidelines: http://journals.plos.org/plosone/s/licenses-and-copyright.

1. You may seek permission from the original copyright holder of Figure 3 to publish the content specifically under the CC BY 4.0 license.  

Reviewers' comments:

Reviewer's Responses to Questions

**Comments to the Author**

1. Is the manuscript technically sound, and do the data support the conclusions?

Reviewer #1: Yes

2. Has the statistical analysis been performed appropriately and rigorously? 

Reviewer #1: Yes

3. Have the authors made all data underlying the findings in their manuscript fully available?

Reviewer #1: Yes

4. Is the manuscript presented in an intelligible fashion and written in standard English?

Reviewer #1: Yes

5. Review Comments to the Author

Reviewer #1: This paper looks like a report rather a scientific paper. Authors should have consulted authors' guidelines carefully before submission.

Another issue lies with novelty, no clear hypothesis or research questions to test.

Many subsections are unorthodox. I do not know why need to add such headings rather follow a standard pattern.

Positive externalities are given, how about negative?

Revise Figure 3 for accuracy and better map/location/resolution, etc.

And few other issues such as report style write up and settings.

Please revise all carefully and then submit.

6. PLOS authors have the option to publish the peer review history of their article (what does this mean?). If published, this will include your full peer review and any attached files.

Reviewer #1: No

---

## [Author Response · Author response to Decision Letter 0]

14 Sep 2022

Response to Editor

Dear Associate Professor Ghaffar Ali

 Thank you very much for giving us the chance to revise and resubmit the manuscript. We truly appreciate the valuable comments and suggestions that have helped substantially improve the quality of the paper.

 Based on the submission guidelines and map copyright guidelines you sent to us, we carefully changed the style form of this manuscript to meet the journal requirements: 1. Moved the content of footnotes into the main text and deleted footnotes; 2. Changed the Reference list into “Vancouver” style; 3. Made the line spacing, abbreviations as well as equations meet the style requirments, etc. We also rebuilt Fig 3 by using the map and data from OpenStreetMap site that are compatible with CC BY licensing and added the map statement in Acknowledge. The new Fig 3 is as follows:

Note: Base map and data from OpenStreetMap and OpenStreetMap Foundation, original copyright: https://www.openstreetmap.org/copyright. The stars mark the locations of air monitoring stations, and the dots mark the locations of domestic waste incineration plants in Shanghai.

We have uploaded figure files to the Preflight Analysis and Conversion Engine (PACE) digital diagnostic tool, https://pacev2.apexcovantage.com/. 

Following the Reviewer’s comments, we have made the following major revisions to the paper. Note that we would not like to present some minor revisions to minimize the length of the letter.

1. “This paper looks like a report rather a scientific paper. Authors should have consulted authors' guidelines carefully before submission.“

We have carefully changed the style form of this manuscript to meet the journal requirements. We have reorganized the structure of this manuscript and rewritten the introduction section by directly stating our research question and addressing how our work contributes to the literature.

Specifically, the revised chapter structure is:

1.Introduction

2.Materials and Methods

 2.1 Policy Background and Research Hypotheses 

 2.2 Data Source and Description

 2.3 Empirical Strategy 

3.Results

 3.1 Baseline Findings

 3.2 Parallel Trend Test

 3.3 Robustness Tests

 3.4 Spillover Effect

4.Discussion

5. Conclusions

Acknowledgments

References

2. “Another issue lies with novelty, no clear hypothesis or research questions to test.”

This comment is very enlightening. We have improved the manuscript in two ways. First, in the Introduction section, we directly state our research question after providing background. Then, we address how our work contributes to the literature. Second, we reorganized the structure of this manuscript, adding a subsection: 2.1 Policy Background and Research Hypotheses to clearly propose the three research hypotheses in this study.

Specifically, the research questions are:

Therefore, in this study, we focus on the questions of whether the MSW sorting policy in Shanghai significantly improves urban air quality and how large the influence will be.

The three research hypotheses are:

H1: The MSW sorting policy in Shanghai will reduce the air pollutants generated by waste incineration plants and significantly improve the air quality in Shanghai.

H2: The degree of influence on air quality will be affected by the distances to waste incineration plants. The closer to waste incineration plants, the larger the influence will be.

H3: There will be a positive spillover effect on the air quality in adjacent cities.

3.“Many subsections are unorthodox. I do not know why need to add such headings rather follow a standard pattern.“

We have reorganized the structure of this manuscript and made each heading the same as the requirements of the Manuscript Organization in journal guidelines.

Specifically, the main modifications include:

Changed the “2. Policy Background” into a subsection in 2.Materials and Methods “2.1 Policy Background and Research Hypotheses”;

Changed the “4.2 Common Trend Test” into “Parallel Trend Test”;

Changed the “6. Conclusion and Policy Implications” into “5. Conclusions”

4.“Positive externalities are given, how about negative?“

We have added a new paragraph in the Discussion section to address the immature aspects of the MSW sorting policy in Shanghai, which has caused inconvenience to residents.

Finally, there are still many immature aspects of the MSW sorting policy in Shanghai, which has caused inconvenience to residents. Wu et al. (2021) [34] used a text mining technique to uncover attitudes from the Chinese public toward MSW sorting policies. They find that although a large proportion of the Chinese public has a positive attitude toward the MSW sorting policy, the proportion of people with negative emotions reached nearly half. Most negative emotions were toward fines, MSW sorting rules, fees, timing of throwing waste, and irregular recycling procedures. Our parallel trend test also shows that the impact of the MSW sorting policy seems to become weaker in the long term. Therefore, to maintain the sustainability of MSW sorting behaviors, the government needs to further improve efficiency and reduce residents' time costs.

5.“Revise Figure 3 for accuracy and better map/location/resolution, etc. In addition, few other issues such as report style write up and settings.“

We have rebuilt Fig 3 as above. We also avoided the writing style of the report by removing some unnecessary statistics listed in the Introduction section, removing the policy implications in the Conclusion section and using an editing tool to improve the write up of this manuscript.

---

## [Decision Letter · Decision Letter 1]

19 Oct 2022

The Impact of Municipal solid waste sorting policy on Air Pollution: Evidence from Shanghai, China

PONE-D-22-13042R1

Dear Dr. Wang,

We’re pleased to inform you that your manuscript has been judged scientifically suitable for publication and will be formally accepted for publication once it meets all outstanding technical requirements.

Kind regards,

Ghaffar Ali, PhD

Academic Editor

PLOS ONE

Additional Editor Comments (optional):

Reviewers' comments:

Reviewer's Responses to Questions

**Comments to the Author**

1. If the authors have adequately addressed your comments raised in a previous round of review and you feel that this manuscript is now acceptable for publication, you may indicate that here to bypass the “Comments to the Author” section, enter your conflict of interest statement in the “Confidential to Editor” section, and submit your "Accept" recommendation.

Reviewer #1: All comments have been addressed

2. Is the manuscript technically sound, and do the data support the conclusions?

Reviewer #1: Yes

3. Has the statistical analysis been performed appropriately and rigorously? 

Reviewer #1: Yes

4. Have the authors made all data underlying the findings in their manuscript fully available?

Reviewer #1: Yes

5. Is the manuscript presented in an intelligible fashion and written in standard English?

Reviewer #1: Yes

6. Review Comments to the Author

Reviewer #1: Authors have made changes in the light of comments and suggestions given earlier. I am satisfied with the revision.

7. PLOS authors have the option to publish the peer review history of their article (what does this mean?). If published, this will include your full peer review and any attached files.

Reviewer #1: No

---

## [Editor Report · Acceptance letter]

24 Oct 2022

PONE-D-22-13042R1 

The Impact of Municipal solid waste sorting policy on Air Pollution: Evidence from Shanghai, China 

Dear Dr. Wang:

I'm pleased to inform you that your manuscript has been deemed suitable for publication in PLOS ONE. Congratulations! Your manuscript is now with our production department. 

Kind regards, 

on behalf of

Prof. Ghaffar Ali 

Academic Editor

PLOS ONE